# EFFICIENT THINKING VIA META CHAIN-OF-THOUGHT EVALUATION

## ABSTRACT

Large Language Models (LLMs) have shown remarkable capabilities in handling complex tasks. Recent breakthroughs in Large Reasoning Models (LRMs)such as OpenAIs o1 and DeepSeek-R1have pushed performance even further in System-2 reasoning domains like mathematics and programming. By leveraging supervised fine-tuning (SFT) and reinforcement learning (RL), these models enhance Chain-of-Thought (CoT) reasoning. However, while longer CoT sequences boost accuracy, they also lead to increased computational costs due to verbose and redundant outputs, a challenge termed the "overthinking phenomenon". In this paper, we design a novel and efficient framework called Dynamic Verify Stopping in Long Reasoning (DVS-LR) to resolve the issue of overthinking. An early-stop verifier is trained to evaluate the Meta-CoT from multiple dimensions like completeness, correctness, and self-validation. DVS-LR receives the generating CoT stream and activates the verifier at adaptive checkpoints. If the score received from the verifier reaches a threshold, the current CoT generation is terminated, and the LRMs directly outputs the final answer without further thinking. Experiments on various math tasks benchmark show that our proposed method achieves 30% cut ratio while maintaining original accuracy. Based on the observation of the average length of after-cut length, we propose the "Budget Forcing Early Stop Majority Voting" method when the token budget is fixed. Experiments show that this method can improve the accuracy on various benchmark compared with original one chain generation.

## 1 INTRODUCTION

Large Language Models (LLMs) have become powerful AI tools, excelling in natural language understanding and complex reasoning. Recent advancements in reasoning-focused LLMsoften termed Large Reasoning Models (LRMs) (e.g., OpenAIs o1 and DeepSeek-R1)have further enhanced performance in System-2 reasoning tasks, particularly in mathematics and programming (Jaech et al., 2024; Guo et al., 2025). Built upon pretrained foundational models optimized for next-token prediction, these LRMs employ Chain-of-Thought (CoT) reasoning, generating explicit, step-by-step logical sequences before delivering final answers. This approach significantly boosts their ability to handle reasoning-intensive challenges.

However, the generation of overlong CoT sequences also significantly increases computational over road and reasoning latency, which hinders their deployment in computationally sensitive real-life applications. Moreover, recent research reveals an intrinsic overthinking problem in LRMs: These models persistently generate verbose reasoning sequences, introducing irrelevant information and unnecessary thought steps (Sui et al., 2025). Such redundant processing not only wastes computational resources but also leads to accuracy degradation by derailing from correct reasoning paths to erroneous ones. This redundancy can be attributed to the design of the supervised fine-tuning or Reinforcement Learning stage, where the ability to dynamically adjust its reasoning length during generation is overlooked, leaving a gap in the inference efficiency of LRMs.

Intuitively, the reasoning process generated by LRMs follows from step to step. This process is termed as "Meta-CoT" (Xiang et al., 2025). Some logical transitions exist to help model think from other directions. Those transition step often starts with word like "Wait", and "Alternatively" (Lu et al., 2025). This makes the LRMs have the ability to explore more possible solutions, but it will

also lead to he "overthinking" as the LRMs might generate the answer in early step but still keep thinking more solutions. To resolve this issue, an intuitive idea is to stop the reasoning process timely when LRMs make sure its solution is correct, and then force the model to generate the output directly without any further thinking (Yang et al., 2025). Then a natural question raises:

*How do we confirm that the current CoT is sufficient to generate the correct answer?*

In this paper, we propose a novel framework called Dynamic Verify Stopping in Long Reasoning (DVS-LR) to resolve the issue of overthinking. First, an early stopping verifier is trained by supervised finetuned on a variable-length Meta-CoT dataset. Specifically, the dataset is constructed by splitting a long CoT generated by DeepSeek-R1 into multiple traces based on logical transition. The verifier will output a justification score for the given Meta-CoT based on completeness, correctness and existence of verification. Then, the LRMs generates the CoT stream and the algorithm calls verifier at each time the CoT comes up with a solution or logical transition. If the verifier generates a score exceeding a threshold. A stop thinking tag will be added to the end of current CoT and the LRMs will directly output the answer without further thinking.

Based on the observation that the token length of early stop chain is mostly upper bounded, we propose the "Budget-Forcing Early Stop Majority Voting" method when total token budget is limited. For example, when token budget is 32K, we turn to generate 4 parallel chains with each token length bounded by 8K. DVS-LR framework is also implemented for each chain's generation. The final answer is generated by running majority voting algorithm over 4 outputs. Compared with original one chain generation with 32K budget, our method utilize the token more efficiently and the majority voting improves the answer's accuracy.

Our method is simple yet effective, and can be seamlessly extended to different sizes of reasoning models, achieving excellent results in the six most popular math benchmarks, including AIME 2025, AIME 2024, AMC 2023, Olympiad Bench, and MATH-500. We test the effectiveness and efficiency of our proposed DVS-LR frame work. Experimental results show that our method can reduce the average token length by $30.7\%$, while the overall accuracy increases $0.4\%$. This shows that our method can effectively and correctly identifies those chains generating answer early, avoiding overthinking phenomenon. Additionally, our "Budget-Forcing Early Stop Majority Voting" method can significantly increase the overall accuracy from $73.8\%$ to $76.7\%$, which demonstrates the effect of multi-chain generation with majority voting.

## 2 RELATED WORK

In this section, we summarize the related work on efficient reasoning. By different stages in CoT generation, the efficient reasoning approach can be classified by three types: training time model-based efficient reasoning, test time output-based efficient reasoning, and input prompt-based efficient reasoning.

### 2.1 MODEL-BASED EFFICIENT REASONING

Model-based efficient reasoning method considers optimizing full-length reasoning models into more concise reasoning models or directly training efficient reasoning models. First, some studies try to implement RL optimization via length reward (Luo et al., 2025; Aggarwal & Welleck, 2025; Arora & Zanette, 2025). These works propose integrating a length reward into the RL framework,. In principle, the length reward assigns higher scores to short, correct answers while penalizing lengthy or incorrect ones, thereby optimizing the length of the reasoning path. Other studies train SFT with variable-length CoT (Munkhbat et al., 2025; Xia et al., 2025; Kang et al., 2025; Han et al., 2024; Lu et al., 2025). Variable-length CoT reasoning datasets refer to datasets of long/short reasoning steps that could guide LLMs to achieve correct answers.

### 2.2 INPUT PROMPTS-BASED EFFICIENT REASONING

Input prompts-based efficient reasoning method seeks to enhance reasoning efficiency based on input prompt properties such as difficulty or length control. Prompt-guided efficient reasoning explicitly instructs LLMs to generate fewer reasoning steps, can be a straightforward and highly effective

method for improving the efficiency of reasoning models (Xu et al., 2025; Lee et al., 2025). Some works consider prompts attribute-driven reasoning routing, where routing strategies for efficient reasoning dynamically determine how language models handle queries based on their complexity and uncertainty (Aytes et al., 2025; Chuang et al., 2025).

### 2.3 REASONING OUTPUT-BASED EFFICIENT REASONING

Reasoning output-based efficient reasoning method aims to dynamically reduce reasoning steps and length during inference. Some studies utilize latent representation compression (Hao et al., 2024; Shen et al., 2025b; Cheng & Van Durme, 2024; Shen et al., 2025a). These methods can be categorized into two types: training LLMs to inference using latent representations or using an auxiliary model. Other works study the dynamic reasoning paradigm (Sun et al., 2024; Wang et al., 2025; Ding et al., 2025). Current training-free approaches explore dynamic reasoning using various criteria, such as reward-guided, confidence-based, and consistency-based selective reasoning.

Some works study the early stop approach like this paper, where the LRMs is forced to stop thinking when some criteria is reached (Yang et al., 2025; Pu et al., 2025). Pu et al. (2025) considers first estimates the token budget needed for each problem, and then it monitors the CoT generations, terminates when it reaches the estimated budget. However, this method can not adaptively check the current chain, leading to the possibility of incomplete reasoning. In contrast, our method verifies chain streamly and stop when it gets high score. Yang et al. (2025) implements the similar idea of dynamic early stopping, while they stop the chain when the sequence output probability is larger than some threshold, indicating LRM is somewhat confident about its generation. Unlike their implicit verification, we explicitly use a verifier to score the chain from multiple dimensions, leading to a more interpretable evaluation criteria.

## 3 PRELIMINARIES

### 3.1 GENERATION OF LARGE REASONING MODEL AND META CHAIN-OF-THOUGHT

In contrast to traditional large language models (System 1), large reasoning models (System2) exhibit distinct generation patterns during the inference stage. (1) LRMs use tags to divide the output into two processes: slow thinking and conclusion. LRMs conduct systematic and thorough reasoning in the slow thinking, ultimately summarizing the thought process and providing the final answer in the conclusion. (2) During the slow thinking process, LRMs engage in complex thinking actions, such as Problem Restatement and Comprehension, Approach Exploration, and Result Verification.

Meta-CoT framework decomposes the CoT into multiple thinking step. We refer to each thinking action as a thinking step $s$, and the transitions between these chunks are often marked by action transition points, including Wait, Alternatively, and Hmm. Suppose for a given prompt $x$, the LRM generates total $N$ thinking steps $\{s_i\}_{[N]}$. The output $y$ of an LRM can be formulated as "<think> + $\sum_{i=1}^{N} s_i$ + </think> + $o$", where $o$ is the final output after thinking shown to the user.

### 3.2 REASONING STRUCTURE OF META CHAIN-OF-THOUGHT

Each Meta-CoT trajectory $C$ can be decomposed in to a set of thinking steps $\{s_i\}_{i\in[N]}$. Each denotes an individual thought, and each thought may perform distinctive role such as trying out a new solution strategy, reflecting its progress, back-tracking or verifying calculations, etc. In order to differentiate between independent thinking steps, we attend to the fact that models often leverage transition keywords to make a natural transition between thoughts. We utilize these linguistic markers to segment and extract individual thinking steps from the full reasoning trace.

The over-thinking issue: too many redundant thoughts. LRM may expend excessive compute on questions that are exceptionally simple or for which the answer is already evident. The model tends to generate unnecessary thoughts such as self-doubt and redundant verification, even when it produces the correct answer within its early steps. In over-thinking, despite the answer being evident, a new thought is started instead of directly generating the answer. From Figure 1, it can be seen that even though LRM has generated the correct answer (pink node), it still continues thinking process by

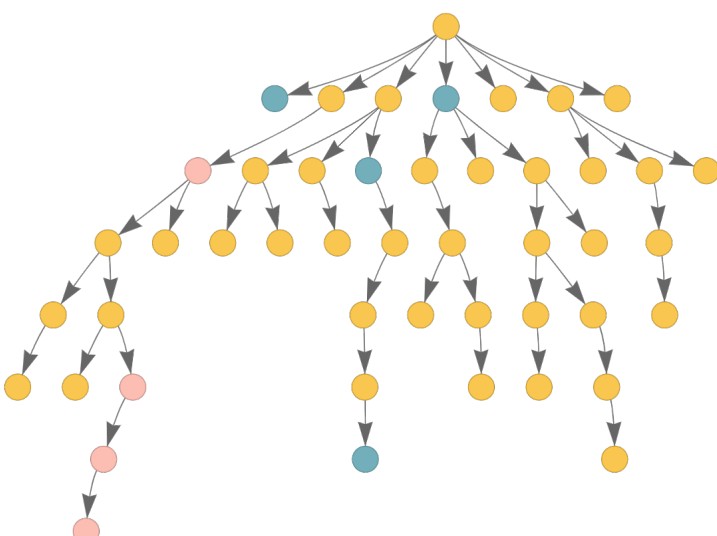

Figure 1: An thinking tree structure for CoT generated by DeepSeek-R1. The yellow node denotes the normal thinking step, the blue node denotes the verify step where the LRM tries to check the correctness of its thinking, and pink node denotes the step where LRM generates the correct answer. The directed edge denotes the logical relation from in-node to out-node.

performing multiple verifications (blue node) and new direction thought (new branch in the tree). These steps are actually unnecessary in the thinking generation.

## 3.3 BUDGET FORCING

The budget forcing methodology capitalizes on the intrinsic reasoning dynamics of Large Reasoning Models (LRMs) to regulate computational expenditure during inference, achieving efficiency gains without architectural modifications or retraining ((Muennighoff et al., 2025)). Specifically, upon reaching the maximum token count, they appended the end-of-thinking token "</think>" along with Final Answer: to early exit the thinking stage. To enforce a minimum, they suppressed the generation of the end-of-thinking token delimiter and appended action transition points to the LRMs ongoing thought process, thus encouraging the model to double-check its answer or attempt new reasoning approaches.

## 4 METHOD

### 4.1 CONSTRUCT REASONING TREE OF CHAIN-OF-THOUGHT

To construct the thinking tree structure for a given Meta-CoT, we analyze the reasoning process step-by-step. Typically, when solving problems, LLMs generate CoT sequences where logically connected steps begin with transition words like "Thus,," "Therefore," or "So," indicating a continuation of the same reasoning path. However, the reasoning process is not strictly linearit often branches when encountering alternative thinking chunks triggered by words like "Wait," "Alternatively," or "Hmm." These transition words signal shifts in reasoning direction, requiring the model to trace back and connect to a relevant prior step. To systematically capture this branching logic, we propose the Extracting Tree Structure algorithm, which identifies these transitions and reconstructs the hierarchical reasoning flow by establishing connections between related steps while preserving the non-linear nature of human-like problem-solving.

To construct a CoT reasoning tree, the algorithm begins by splitting the full reasoning chain into discrete steps using step-starting words (e.g., "First", "Then", "Therefore"), with each step representing a node in the tree. It then analyzes the logical relationships between steps, where transition

words (e.g., "Wait", "Alternatively") signal a shift to a new thinking chunk, prompting a branch switch in the tree structure. When the model switches reasoning paths, it traces back to a prior step that aligns with the new line of thought, identified by applying the Longest Common Subsequence (LCS) algorithm to locate the most textually similar historical stepindicating logical continuity. The algorithm establishes a directed edge between this transition node and the current step, iteratively building the hierarchical tree by connecting all steps. This approach dynamically captures branching logic, dependencies, and reasoning flow while mitigating redundancy in multi-step inference. The algorithm detail can be found in Appendix B.

*Remark* 4.1. In this algorithm we apply the LCS algorithm to find the transition node in the tree. This is because we observe that LRMs often restate the prior thinking when it tries to explore other directions. LCS algorithm helps to identify which prior node contains this restatement, which indicates that current node is likely to continue thinking from that prior node's thinking step. We note that LCS algorithm may be replaced by other text matching algorithm, or we can also call the LLMs to help us find the transition node.

## 4.2 TRAINING EARLY STOP VERIFIER

To construct the training dataset for the early-stop verifier via SFT, we generate variable-length Meta-CoT sequences using reasoning trajectories from the S1K math problem dataset (Muennighoff et al., 2025). For each CoT trajectory, we first apply the Extracting Tree Structure algorithm to build its hierarchical thinking tree. Each leaf node in this tree represents the termination point of a distinct reasoning chunk. By truncating the original CoT at these leaf nodes, we systematically create early-stopped CoT variants. This decomposition process enables each complete DeepSeek-R1 CoT trajectory to yield multiple partial reasoning paths of varying lengths, forming a diverse SFT dataset that captures different stages of the reasoning process while preserving logical coherence.

After generating variable-length Meta-CoT sequences, we assess their generation qualities. This evaluation framework rates each CoT across six dimensions: (1) Subtask Coverage, (2) Critical Steps Covered, (3) Final Answer Derived, (4) Mathematical Validity, (5) Final Answer Accuracy, and (6) Self-Validation. Each criterion is scored on a 0 - 10 scale, with the verifier providing structured justifications in the format "Dimension: Reason + Score". This dual-output reward system delivers not only scalar quality metrics but also detailed rationales for each rating, thereby improving the interpretability and refinement of the reasoning process (Liu et al., 2025). The combination of quantitative scores and qualitative feedback enhances both the evaluation transparency and the subsequent CoT optimization.

We train the early-stop verifier on this dataset using existing pretrained LLMs. The verifier evaluates variable-length Meta-CoT sequences and generates detailed justifications for its assessments. Each CoT receives a composite score ranging from 0 to 60, derived from six distinct evaluation dimensions (each scored 0-10). This multidimensional scoring system provides both quantitative ratings and qualitative feedback, enabling comprehensive analysis of reasoning quality at any stopping point in the CoT sequence. The details of the verifier can be found in Appendix A.

## 4.3 DYNAMIC VERIFY STOPPING IN LONG REASONING FRAMEWORK

In this subsection, we introduce our proposed Dynamic Verify Stopping in Long Reasoning (DVS-LR) framework that helps LRMs generates more efficient CoT.

The Large Reasoning Model (LRM) generates a continuous CoT stream, with the algorithm invoking the verifier whenever the CoT either produces a potential solution or undergoes a logical transition (marked by specific transition words). The early-stop verifier evaluates these checkpoints and assigns a comprehensive score from 0 to 60. When this score surpasses a predetermined quality threshold, the system determines that the current CoT has achieved sufficient reasoning quality. At this point, a stop thinking "</think>" tag is appended to terminate further reasoning, and the LRM immediately outputs the final answer without additional processing steps.

## 4.4 BUDGET FORCING EARLY-STOP VERIFIER MAJORITY VOTING

Based on our observation of those early stop chain, the average token length of CoT after-cut is about 8K, which means majority problems can be solved within 8K token budget. Thus when the token

---

**Algorithm 1** Dynamic Verify Stopping in Long Reasoning Framework

---

**Require:** Query $Q$, threshold $\lambda$.
**Ensure:** Final answer with terminated CoT
1:  Initialize CoT stream $\leftarrow \emptyset$
2:  $stop\_thinking \leftarrow$ False
3:  Append "<think>" token to CoT
4:  **while** not $stop\_thinking$ **do**
5:      Generate next reasoning step $s$ using LRM(Q).
6:      Append $s$ to CoT stream
7:      **if** $s$ contains *(potential solution)* **then**
8:          $checkpoint \leftarrow$ current CoT state
9:          $score \leftarrow$ VERIFIER($checkpoint$)
10:         **if** $score \geq \lambda$ **then**
11:             $stop\_thinking \leftarrow$ True
12:             Append "</think>" to CoT
13:         **end if**
14:     **end if**
15: **end while**
16: Extract final answer $a$ from terminated CoT
17: **return** Formatted answer $a$ with terminated CoT

---

budget is fixed (like 32K, the classic max token upper bound set by existing LRM.), unlike traditional one-time long chain thinking, it is more likely to output the correct answer by generating 4 chains with max length of 8k.

The algorithm calls an LRM 4 times to generate answer parallelly. For each chain, the algorithm still performs our DVS-LR framework to verify if current chain can be terminated. If the length of the chain reaches budget, it will be terminated. Then the algorithm gathers 4 outputs from LRMs, and it will run the majority voting algorithm to generate the final solution.

## 5 EXPERIMENTS

### 5.1 EXPERIMENTAL SETUP

We evaluate models on seven math-specific benchmarks: AIME25, AIME24, AMC23, Olympiad-Bench, and MATH500. The first four benchmarks focus on olympiad-level math problems, where AIME25 and AIME24 both comprise 30 challenge problems selected from the 2025 and 2024 American Invitational Mathematics Examination (AIME); AMC23 contains 40 mathematical problems proposed by American Mathematics Competitions (AMC) at 2023; OlympiadBench contains 674 problems. MATH500 includes 500 high-school math competition problems.

For evaluation, we report accuracy, average length, and cut ratio to measure the performance of DVS-LR framework and Budget-Forcing Dynamic Verify Majority Voting algorithm. For accuracy, we use exact match between the models prediction and the reference answer. Formally, the accuracy is computed as $\frac{1}{N}\sum_{i=1}^{N}\mathbb{I}\{\text{LRM}(x_i) = a_i\}$, where $\{(x_i, a_i)\}_{[N]}$ is the set of $N$ data with problem $x_i$ and answer $a_i$, and $\text{LRM}(x_i)$ is the output solution by LRM given problem $x_i$. The ground-truth answers to the evaluation problems in our experiments are all well-structured numerical values or options. Therefore, we apply rule-based evaluations directly to verify mathematical equivalence. For average length, we first tokenize the CoT and compute the number of output tokens. Suppose $L_i$ is the token length of CoT $C_i$, and $L_i'$ is the token length of applying DVS-LR method on CoT $C_i$. Then we take average over $N$ CoT lengths and get the average length. For cut ratio, it is defined as the ratio of early stopped CoT, i.e., $\frac{\sum_{i=1}^{N}\mathbb{I}\{L_i' < L_i\}}{N}$. This measures the efficiency of the early stop verifier.

Our early stop verifier is trained on the variable length CoT evaluation datasets mentioned before in Sec 4.2. To be specific, the original CoT dataset is collected from DeepSeek-R1 by solving the s1K math problem dataset (Muennighoff et al., 2025). The evaluation text is collected by calling Gemini-

**Algorithm 2** Budget Forcing Early Stop Majority Voting

**Require:** Query $Q$, score threshold $\lambda$, max token budget $L_{max}$, number of parallel runs $T$.
**Ensure:** Final answer $A_{final}$
1: Initialize answer set $\mathcal{A} \leftarrow \emptyset$
2: Initialize token budgets $L[c] \leftarrow L_{\max}, c \in [T]$
3: **for** chain $c = 1$ to $T$ **in parallel do**
4:     $C_c \leftarrow \text{LRM}(Q)$
5:     **while** $L[c] > 0$ **do**
6:         Generate next step $s \leftarrow \text{LRM}(C_c)$
7:         Append $s$ to $C_c$
8:         $L[c] \leftarrow L[c] -$ token length of $s$
9:         **if** $s$ contains *(solution)* **then**
10:            $checkpoint \leftarrow C_c$
11:            $score \leftarrow \text{VERIFIER}(checkpoint)$
12:            **if** $score \geq \lambda$ **then**
13:                Append "</think>" to $C_c$
14:                **break**
15:            **end if**
16:         **end if**
17:     **end while**
18:     $\mathcal{A} \leftarrow \mathcal{A} \cup \{\text{EXTRACTANSWER}(C_c)\}$
19: **end for**
20: $A_{final} \leftarrow \text{MAJORITYVOTE}(\mathcal{A})$
21: **return** $A_{final}$
22: **procedure** MAJORITYVOTE($\mathcal{A}$)
23:     Count occurrences for each unique answer
24:     **return** answer with max count (ties broken randomly)
25: **end procedure**

2.5-flash model to formally evaluate and scoring the CoT from six dimensions. The benchmark math problems are processed by DeepSeek-R1-Distill Qwen-32B model (Guo et al., 2025).

## 5.2 EVALUATION RESULTS

We first test the effectiveness and efficiency of our DVS-LR framework with the classic LRM generations. In Table 1 we report the accuracies and average lengths of two methods. Besides, we also count the cut ratio and truncation ratio to show how often the early stop works and how long it cuts the original CoT.

From Table 1 it can be seen that the DVS-LR method has the similar accuracies with original full chain, which implies that our early stop verifier can provide the stop signal correctly, allowing the LRM have adequate thinking steps to generate the final answer. The comparisons over the average length between two methods show that the DVS-LR framework can efficiently truncate about 30% tokens of original CoT, which mostly includes excessive validation and over exploration for other solutions. Overall cut ratio shows that nearly 70% chains can be determined to stop by the verifier.

Additionally, the early stop verifier shows different cut performance on different dataset. For simpler math datasets like MATH500 and AMC23, for which original CoT has achieved high accuracies, the DVS-LR method can significantly cut $80\% - 90\%$ chains, achieving truncation ratio about 40%. For the harder datasets like AIME25/24 and OlympiadBench, the DVS-LR method performs more conservative, cutting less chains and resulting in less truncation ratio. This result shows that our method performs adaptively with the difficulty of the problem, as it will give LRM more thinking steps facing hard problems and push the LRM quickly derive the output facing easy problems. This aligns with our intuition and desire property.

Then we provide the performance of our budget forcing early stop majority voting algorithm. We report the accuracies in Table 2 over benchmarks of "One Chain with 32K Budget Forcing" (32K

| Dataset | Cut ACC | Origin ACC | Cut Legnth | Origin Length | Cut Ratio |
|---------|---------|-----------|------------|---------------|-----------|
| MATH500 | 95.6% | 96.0% | 3818 (-42.9%) | 6817 | 92.4% |
| AIME25 | 46.7% | 46.7% | 10295 (-22.3%) | 13257 | 40.0% |
| AIME24 | 60.0% | 60.0% | 9456 (-30.1%) | 13134 | 56.7% |
| AMC23 | 92.5% | 90.0% | 6124 (-38.4%) | 9946 | 82.5% |
| OlympiadBench | 76.3% | 76.4% | 7452 (-29.1%) | 10504 | 67.8% |
| Overall | 74.2% | 73.8% | 7429 (-30.7%) | 10731 | 67.9% |

Table 1: Performance comparison with DVS-LR and original chain with no early stop on the open-source DeepSeek-R1-Distill 32B model. We report the accuracy of DVS-LR (Cut ACC), accuracy of original chain (Origin ACC), average length of DVS-LR (Cut Length), average length of original chain (Origin Length), cut ratio (Cut Ratio).

BF), "8K*4 Majority Voting" method (8K*4 BF), and "8K*4 Early Stop Majority Voting" method (8K*4 Early Stop).

From Table 2, we can see that our "8K*4 Early Stop Majority Voting" method outperforms than other two methods on all math datasets. Additionally, "8K*4 Majority Voting" method also outperforms "One Chain with 32K Budget Forcing" method. This observation implies that multiple shorter thinking may be more likely to reach the correct answer than one longer thinking.

To illustrate this phenomenon, we can trace the logical relations over thinking steps in one chain. By extracting the tree structure of one chain, we can find that the width of the tree is large but the depth is moderate (Figure 1). This observation indicates that when LRM generates thinking, it frequently changes its thinking step to other directions, resulting a new branch in the thinking tree. Thus when more tokens are used, it may not help LRM digs deeper in one direction, but tries multiple solution directions to reach the correct one. When we turn to generate multiple parallel thinking chains, it will 4 thinking trees correspondingly. Unlike one chain generation, 4 thinking trees might start thinking from different perspective and make transitions more diverse by the output randomness. Thus it is more likely to reach the correct thinking path. However, note that sufficient token budget is also required since it guarantees the thinking depth of one thinking path. The balance of thinking depth and width is the key to generating the final correct answer.

| Method | MATH500 | AIME25 | AIME24 | AMC23 | OlympiadBench | Overall |
|--------|---------|--------|--------|-------|---------------|---------|
| BF | 96.0 | 46.7 | 60.0 | 90.0 | 76.4 | 73.8 |
| BF + MV | 96.0 | 49.2 | 63.3 | 95.0 | 76.4 | 76.0 |
| BF + MV + ES | **96.2** | **50.0** | **65.0** | **95.0** | **77.2** | **76.7** |

Table 2: Accuracies comparisons with one chain 32K budget, 8K*4 Majority Voting, 8K*4 Early Stop Majority Voting methods on the open-source DeepSeek-R1-Distill 32B model.

The early stop performance of "8K*4 Early Stop Majority Voting" is shown in Figure 2. As we can see, the harder the problem is, the less the chains are cut. It matches the result shown in Table 1. Additionally, we also report the ratio of reaching max token budget (8K) on benchmarks. It can be seen that only few chains reach the budget with the early stop verifier, which aligns with the average length result shown in Table 1. This shows that our DVS-LR framework helps LRM generates full answer before reaching budget limit, leading to more accuracies.

We also test the performance of DVS-LR framework with different verifier threshold. Specifically, "8K*4 Early Stop Majority Voting" method is tested with threshold $\lambda$: 50, 55, 60, shown in Figure 3. It can be seen that with larger threshold, the more accurate the answer is, and the less chains are cut. For the reach max token budget ratio, it increases with the threshold increasing. This is because the larger threshold is, the chain is harder to be cut, and more chains are likely to reach the max token budget.

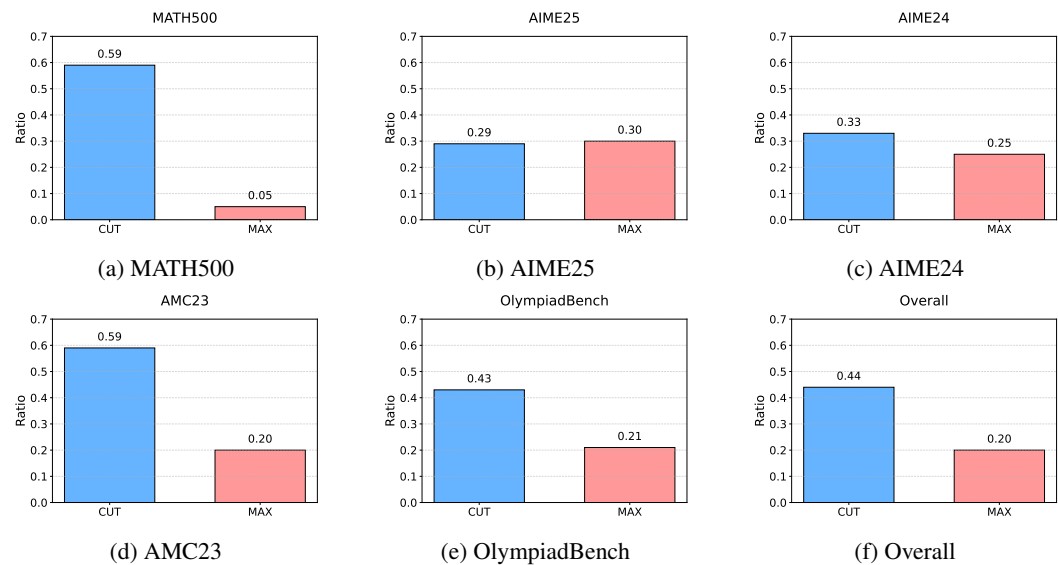

Figure 2: Performance of cut ratio (CUT) and reach budget ratio (MAX) for 8K * 4 early stop majority voting algorithm over benchmarks.

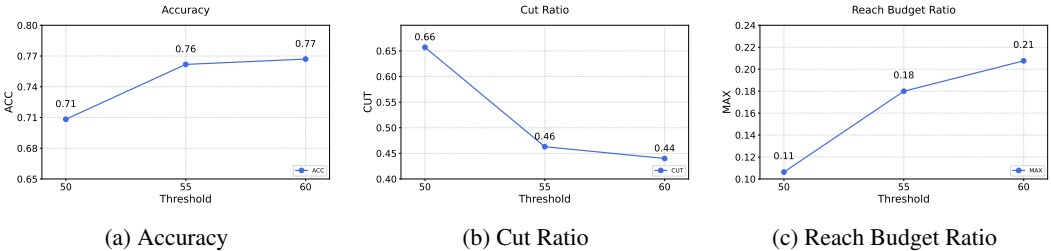

Figure 3: Performance of accuracy (ACC), cut ratio (CUT), and reach budget ratio (MAX) for 8K*4 Early Stop Majority Voting algorithm over benchmarks with verifier threshold $50, 55, 60$.

## 6 CONCLUSION

This work addresses the "over-thinking" phenomenon in Large Reasoning Models (LRMs) caused by redundant Chain-of-Thought (CoT) sequences. By introducing Dynamic Verify Stopping in Long Reasoning (DVS-LR) framework, we pioneer a lightweight framework that integrates multi-dimensional verification of Meta-CoT to dynamically terminate CoT generation when thresholds are met. Experimental results demonstrate DVS-LR reduces reasoning token counts by 30% across mathematical benchmarks, while preserving baseline accuracy. Furthermore, our Budget Forcing Early Stop Majority Voting method improves accuracy under fixed token budget scenario, outperforming single-chain generation by 2.9%. These innovations establish new paradigms for resource-efficient reasoning in LRMs, enabling deployment in latency-sensitive or compute-limited environments.

Our method only studies the testing-time compute of LRM, which may be constrained by the ability of original model. Future work will explore training-time method that directly optimize the generating process. Specifically, our early stop verifier can be implemented in the variable-length training data generations for SFT. The justified score can also be put into the reward model for RL. Then RL method can be implemented to generate the efficient CoT under the guidance of early stop verifier. This shows the general usage of our verifier. Another direction is the more effective verifier and more efficient triggering checkpoint. Generalist reward modeling method can be applied to the verifier to attain the generality.

**Ethics statement**  This paper presents work whose goal is to advance the field of LLM reasoning. There are many potential societal consequences of our work, none which we feel must be specifically highlighted here.

**Reproducibility statement**  The detail and description of the experiments are provided in experiments and method section.

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

## A  VERIFIER DETAILS

In this section we provide the details of the early stop verifier, including the prompt and output format.

Verifier prompt:

---

**Role Instruction**
Your task is to rigorously evaluate a provided CoT for a math problem using the criteria below.
Rate **each component** on a **010 scale** (10 = fully meets, 5 = partially meets, 0 = fails).
**Evaluation Criteria**

1. **Completeness**

   **Subtask Coverage (010)**: Does the CoT address *every part* of the question (e.g., "Find (x) *and* justify")? 10: All parts addressed. 5: Some parts addressed. 0: No parts addressed.

   **Critical Steps Covered (010)**: Are all necessary steps included (e.g., setup, transformations, calculations)? 10: All steps present. 5: Minor steps missing. 0: Critical steps missing.

   **Final Answer Derived (010)**: Is the answer explicitly stated? 10: Clear final answer. 5: Ambiguous answer. 0: No answer.

2. **Correctness**

   **Mathematical Validity (010)**: Are steps error-free and logically sound? 10: Flawless logic. 5: Minor errors. 0: Major errors/invalid logic.

   **Final Answer Accuracy (010)**: Is the answer numerically/analytically correct? 10: Correct. 0: Incorrect.

3. **Verification Self-Validation (010)**: Does the CoT include checks (e.g., substitution, unit verification)? 10: Full validation. 5: Partial validation. 0: No validation.

**Output Format**
Completeness

- **Subtask Coverage**: [Score]/10, Reason
- **Critical Steps Covered**: [Score]/10, Reason
- **Final Answer Derived**: [Score]/10, Reason

Correctness

- **Mathematical Validity**: [Score]/10, Reason
- **Final Answer Accuracy**: [Score]/10, Reason

Verification

- **Self-Validation**: [Score]/10

Total Score **Total**: [Total] / 60
You will be given a problem and corresponding chain of thought as follows:
**Problem + Input CoT**.

---

Here we provide some output cases of the verifier:

Completeness\n- **Subtask Coverage**: 10/10, The user addresses finding the ratio and the final calculation \(10n + m\).\n- **Critical Steps Covered**: 10/10, The user sets up coordinate systems, derives equations based on geometric constraints, solves the resulting algebraic equations, and calculates the areas and ratio.\n- **Final Answer Derived**: 10/10, The user explicitly calculates the ratio and the final value \(10n + m\).\n\n### Correctness\n- **Mathematical Validity**: 10/10, The derivation of coordinates, the equations from the circle constraint, the algebraic solution for the side length, and the area calculations are all mathematically correct. The handling of the quadratic equation and the selection of the valid solution are sound.\n- **Final Answer Accuracy**: 10/10, The calculated ratio and the final result \(10n + m\) are correct based on the problem statement and the derived values.\n\n### Verification\n- **Self-Validation**: 10/10, The user explicitly checks if the coordinates of points G and H lie on the circle by plugging them into the circle's equation, confirming the validity of the derived side length.\n\n### Total Score\n**Total**: 60/60, The CoT is complete, mathematically sound, and includes self-verification, leading to the correct final answer.

Completeness\n- **Subtask Coverage**: 10/10, The solution addresses all parts of the problem, defining terms, setting up the problem geometrically, and arriving at a numerical answer.\n- **Critical Steps Covered**: 10/10, All necessary steps are included, from setting up the problem geometrically to calculating the final answer.\n- **Final Answer Derived**: 10/10, A clear final answer is stated.\n\n### Correctness\n- **Mathematical Validity**: 5/10, There are errors in the logic. The areas of the four regions are not simply p/2, (1-q)/2, etc. This would only be true if the point X was on an axis. Also, the condition a+c = b+d = 50 is not justified and incorrect.\n- **Final Answer Accuracy**: 0/10, Since there are errors in the derivation, the final answer is incorrect.\n\n### Verification\n- **Self-Validation**: 0/10, There is no validation performed in the CoT.\n\n### Total Score\n**Total**: 35/60, The solution has a good structure and lays out steps, but there are significant mathematical errors.\n"

## B  TREE CONSTRUCTION ALGORITHM

---

**Algorithm 3** Chain-of-Thought Reasoning Tree Construction

---

**Require:** CoT reasoning steps $\mathcal{S} = \{s_1, ..., s_N\}$
**Ensure:** Reasoning tree $T = (V, E)$ with hierarchical dependencies
 1: Initialize tree $T \leftarrow \emptyset$ with root $v_0$
 2: $current\_branch \leftarrow v_0$ ▷ Track active reasoning path
 3: $V \leftarrow [v_0]$ ▷ Sequence of all step nodes
 4: **for** each step $s_i \in \mathcal{S}$ **do**
 5:   Create new node $v_i$ for reasoning step $s_i$
 6:   $V \leftarrow V \cup \{v_i\}$
 7:   **if** $s_i$ starts with step-starting word or logical continue word (e.g., "First", "Then") **then**
 8:     Add edge $(current\_branch, v_i)$ to $E$
 9:     $current\_branch \leftarrow v_i$ ▷ Extend current path
10:   **else if** $r_i$ contains transition word (e.g., "Alternatively", "Wait") **then**
11:     $anchor \leftarrow \text{FINDTRANSITIONANCHOR}(s_i, step\_nodes)$
12:     Add edge $(anchor, v_i)$ to $E$ ▷ Create new branch
13:     $current\_branch \leftarrow v_i$
14:   **end if**
15: **end for**
16: **procedure** FINDTRANSITIONANCHOR($s_i$, $step\_nodes$)
17:   Compute longest common sequence between $s_i$ and all $v_j \in step\_nodes$
18:   $best\_match \leftarrow \underset{v_j}{\arg\max} \text{LCS}(s_i, v_j)$
19:   **return** $best\_match$ ▷ Most semantically similar ancestor
20: **end procedure**
21: **return** $T$ with hierarchical reasoning paths

---

## C    USE OF LARGE LANGUAGE MODEL

This paper did not use LLMs to assist in writing or for other purposes.

