# OpenReview forum: "Efficient Thinking via Meta Chain-of-Thought Evaluation"
_ICLR.cc/2026/Conference — Submitted to ICLR 2026_

### Official Review · Reviewer_weH1 · 2025-10-30

**Soundness:** 2
**Presentation:** 2
**Contribution:** 2
**Rating:** 2
**Confidence:** 4

**Summary:**

The paper proposes an efficient thinking mechanism DVS-LR, a framework to improve reasoning efficiency in Large Reasoning Models by introducing an early-stop verifier that stops generation once it assesses the reasoning trace is already sufficient. The verifier is trained using reasoning trees built from full CoT traces, where each partial CoT (separated in steps) is automatically scored by another LLM on six reasoning-quality dimensions. During inference, the verifier monitors the CoT stream and stops generation when a quality threshold is met. A parallel budget-forcing and majority-voting scheme was proposed to further enhance the accuracy. Experiments show comparable accuracy to full CoT reasoning with shorter outputs during generation.

**Strengths:**

The paper is trying to solve a real-world problem with a long generation in reasoning model.


The empirical results show some improvements that under shorter length, it could reach similar accuracy.

**Weaknesses:**

First, the paper is not well-written. There are a bunch of typos or space related issue starting from introduction section (DeepSeek-R1have, (LRMs)such, LLMsoften, (System 1) vs. System2, Thus„ ,linearit,  stepindicating). Please double check the manuscript to make it organized.


The paper proposes to use a verifier (another LLM) for deciding whether to early stop or not. It needs to evaluate on the checkpoints for each generation segments. Do you have any end-to-end efficiency evaluation regarding including the overhead of the verifier?


Regarding “Extracting Tree Structure algorithm”. The method for constructing verifier training data is not entirely clear to me. Why not simply evaluate the generated CoT step by step, which would be more closely reflecting the inference time flow of reasoning? Instead, the paper restructures full CoTs into reasoning trees and concatenates/reorders nodes for training. Why is this necessary?


The early stop + parallel inference:  The paper says “Based on our observation of those early stop chain, the average token length of CoT after-cut is about 8K, which means majority problems can be solved within 8K token budget.”. Does the generation really reach the maximal of 32k every time with a normal prompt? If generations do not actually reach the 32K token limit, it is unclear how this approach leads to measurable efficiency gains. Moreover, since the length of each reasoning trace is unpredictable beforehand, it is not obvious how one should determine the number of parallel runs to launch in advance.


In the meantime, it is known that this scaling inference[1] like technique will increase the accuracy, with more overhead that it runs several times, the manuscript is currently lacking such results on how much more overhead is induced.


[1] https://huggingface.co/blog/Kseniase/testtimecompute

**Questions:**

See weakness.

---

### Official Review · Reviewer_WYEd · 2025-10-31

**Soundness:** 2
**Presentation:** 3
**Contribution:** 2
**Rating:** 2
**Confidence:** 4

**Summary:**

The paper propose Dynamic Verify Stopping in Long Reasoning (DVS-LR), a method to stop reasoning model overthinking. DVS-LR trains a verifier to evaluate the CoT trajectory, and assign score based on its completeness, correctness (soundness), and self-validation to decide if it should early terminate the reasoning trajectory. Experiment shows a promising token saving in mathematical reasoning tasks.

**Strengths:**

1. Writing. The writing is relatively clear and easy to follow. The illustration helps understand the paper.
2. Methodology. The methodology is relatively sound and practical.

**Weaknesses:**

1. Novelty. Entropy-based early-exit methods (especially training-free methods) has been quite extensively studied in prior works such as the following. The paper needs to make a clearer distinction between the method and the prior works (in the related work). Just listing a few as addition to the current related work section:

- https://arxiv.org/abs/2502.12067
- https://arxiv.org/abs/2412.21187
- https://arxiv.org/abs/2504.01296
- https://arxiv.org/abs/2508.15260
- https://arxiv.org/abs/2412.20993
- https://arxiv.org/abs/2412.18547
- https://arxiv.org/abs/2207.05221

2. Baseline. The paper need to compare against stronger baselines (perhaps one of the above) to demonstrate the trained method indeed has a better token saving compared to the state of the art, while preserving the accuracy.

3. Interpretability and case study. It is unclear that having interpretability as the reward of reasoning trajectory (as proposed in the paper) will have a benefit in boosting the reasoning quality. The paper will benefit by having case study on specific problem (or listing the the graphical representation of the trajectory) to give reader a better understanding.

**Questions:**

1. Novelty and related work as describe in weakness section.
2. (The lack of state of the art) Baselines and experiment as shown in weakness section.
3. Interpretability and case study as describe in weakness.

---

### Official Review · Reviewer_ZVcD · 2025-10-31

**Soundness:** 3
**Presentation:** 3
**Contribution:** 2
**Rating:** 4
**Confidence:** 4

**Summary:**

This paper studies the overthinking and early exit of large reasoning models (LRMs). The authors propose Dynamic Verify Stopping in Long Reasoning, which applies an early stop verifier trained on a Meta-CoT dataset. During answer generation, the verifier justify the completeness, correctness and existence of verification to decide of the thinking should early exit. DVS-LR further applies a total budget based parallel decoding and majority voting. Experiments show that DVS-LR reduces the output length by 30% without sacrificing accuracy.

**Strengths:**

- The Meta-CoT dataset applies LLM-as-a-Judge to improve its scalability.
- The idea of a global budget for the whole CoT with majority voting is novel.

**Weaknesses:**

- This paper lacks comparison with existing works on early exiting language model reasoning, such as [1][2][3]. It is unclear if using Meta-CoT is a more efficient way.
- What is the early stop verifier model architecture, as well as the cost and frequency running the verifier model? If running the verifier model itself has a non-negligible cost, does it reduce the gain of early exiting?

[1] Chen, Xingyu, et al. "Do not think that much for 2+ 3=? on the overthinking of o1-like llms."
[2] Fu, Yichao, et al. "Reasoning without self-doubt: More efficient chain-of-thought through certainty probing."
[3] Zhang, Anqi, et al. "Reasoning Models Know When They're Right: Probing Hidden States for Self-Verification."

**Questions:**

nits:
- L042 "over road" is supposed to be "overhead"?
- L053 "those transition step" should be "steps"?
- L054 "lead to he 'overthinking'" missing "t"
- L056 "when LRMs make sure its solution is correct". "its" should be "their". The same happens in several places (e.g. L065 "the LRMs generates ...")
- L137, Missing hyphens ("System 1" and "System2")
- L372, "the DVS-LR method performs more conservative", should be "conservatively"

---

### Official Review · Reviewer_4hE2 · 2025-11-01

**Soundness:** 2
**Presentation:** 2
**Contribution:** 2
**Rating:** 2
**Confidence:** 4

**Summary:**

This work proposed an early stopping-based method for efficient reasoning.

**Strengths:**

- LRM efficiency is important, and early stopping is a straightforward idea that worth exploring.

**Weaknesses:**

This work is likely too incomprehensive in the evaluation department.

- Lack of model coverage. Seems to only evaluated on one 32B model.
- Lack of dataset coverage. Only evaluated on math reasoning tasks while trained on s1k (also a math dataset).
- No comparison to existing baseline methods.
- No description of basic experiment setting like temperature, number of run, etc.

There should also be better discussion and recognition of tightly connected work. Doing early stopping on CoT/LRM is a well-explored area with literature like HALT-COT, Answer Consistency, FlashThink, DEER... These work are not discussed nor compared. Also, the idea of using "wait" like tokens as separator are also utilized in many reasoning-related work and deserve crediting.

**Questions:**

- Since the proposed method generate 4 chains per question and do majority voting, does the reported length include discarded chains?
- Which model is used as the verifier?

---

### Meta-Review · Area_Chair_6LmN · 2026-01-12

**Summary:**

This paper proposes an inference-time “early stopping” framework for long chain-of-thought reasoning: train a verifier to score partial reasoning traces (completeness/correctness/self-checking), invoke it at checkpoints during generation, and stop once a threshold is met. It also adds a budget-based variant that runs multiple shorter chains with majority voting.

Across reviews, the overall sentiment is negative: the idea is straightforward but the paper does not convincingly establish novelty over existing early-exit / overthinking-mitigation work, and the experimental evidence is not strong enough to support the claimed efficiency gains.

**Reviewer Concerns:**

There is no author rebuttal in the discussion, so the major concerns remain outstanding. The most consistent issues are: (1) insufficient evaluation breadth (single model; math-only tasks; trained and tested within the same narrow domain), (2) missing comparisons to strong existing baselines for efficient reasoning / early exit, and (3) unclear end-to-end efficiency because the verifier itself adds overhead (the paper does not provide a convincing accounting of verifier cost/frequency and how much it offsets token savings).

Reviewers also raised clarity/presentation issues (many typos and missing experimental details like decoding settings), and questioned parts of the methodology (e.g., why the tree-based restructuring is necessary for verifier training vs evaluating step-by-step in the natural generation flow).

**Reviewer Scores:**

N/A as there is no rebuttal..

---

### Decision · Program_Chairs · 2026-01-26

Reject